# Conditional Policy Generator for Dynamic Constraint Satisfaction and Optimization

## Abstract

Leveraging machine learning methods to solve constraint satisfaction problems has shown promising, but they are mostly limited to a static situation where the problem description is completely known and fixed from the beginning. In this work we present a new approach to constraint satisfaction and optimization in dynamically changing environments, particularly when variables in the problem are statistically independent. We frame it as a reinforcement learning problem and introduce a conditional policy generator by borrowing the idea of class conditional generative adversarial networks (GANs). Assuming that the problem includes both static and dynamic constraints, the former are used in a reward formulation to guide the policy training such that it learns to map to a probabilistic distribution of solutions satisfying static constraints from a noise prior, which is similar to a generator in GANs. On the other hand, dynamic constraints in the problem are encoded to different class labels and fed with the input noise. The policy is then simultaneously updated for maximum likelihood of correctly classifying given the dynamic conditions in a supervised manner. We empirically demonstrate a proof-of-principle experiment with a multi-modal constraint satisfaction problem and compare between unconditional and conditional cases.

## 1 Introduction

Constraint satisfaction problems (CSPs) are a fundamental class of combinatorial search problems that arise in numerous domains, including task scheduling, resource allocation, product configuration, and hardware design (Dechter, 1992). They involve finding assignments of values to a set of variables that satisfy a given set of constraints. Many efficient methods to solve them have been long developed, such as backtracking search, constraint propagation, and local search (Kumar, 1992; Rossi et al., 2006). But these algorithms often struggle to scale to complex problems with high dimensionality and diverse multi-modal solutions, and may not generalize well to different problem instances or classes (Dechter, 2003; Kotthoff, 2014; Bengio et al., 2021). In recent years, incorporating machine learning techniques in CSP solvers has emerged as an attractive alternative to address these challenges (Popescu et al., 2022; Sadana et al., 2025). For example, deep neural networks are used to learn search heuristics (Yolcu & Póczos, 2019), to predict the satisfiability of a problem (Xu et al., 2018), to select the best algorithm per case (Loreggia et al., 2016), or to generate solutions directly by learning the optimal solver's decision through imitation or trial-and-error (Bello et al., 2016). Despite the success of these methods, most of them are designed for static CSPs where the problem structure remains unchanged and known a priori. In practice, however, many real-world problems evolve over time or undergo conditional change. For example, in a scheduling problem, a new task may arrive, existing jobs may be delayed, or resource availability may shift, requiring the system to dynamically re-optimize the schedule. Similarly in a car configuration problem, availability of certain features may vary depending on regional regulations or current inventory levels. Therefore, it is crucial to develop CSP solvers that can adapt to these changes in finding solution.

In this work, we propose a reinforcement learning (RL) based method for solving a dynamic CSP (Dechter & Dechter, 1988; Mittal & Falkenhainer, 1990) that especially involves conditional changes in the environment such as conditional activation or deactivation of constraints. We assume that the problem consists of both static and dynamic constraints and variables are independent of each other. By drawing inspiration from

the concept of conditional generative adversarial networks (GANs) (Mirza & Osindero, 2014), we develop a conditional policy generator. By training it through a noise source using the policy gradient method together with the maximum likelihood objective in order to satisfy both static and dynamic constraints, the stochastic policy generator eventually learns the general structure of solution space in a dynamic scenario. In Section 2, background on CSPs, their RL formulation, and related works are provided. Section 3 describes the proposed policy network architecture and training procedure. It is followed by experimental results on a simple multi-modal benchmark problem for both unconditional and conditional cases in Section 4. Finally we discuss future work.

## 2 Background

### 2.1 Constraint satisfaction problems

CSPs are typically represented by a 3-tuple $\langle \mathbf{X}, \mathbf{D}, \mathbf{C} \rangle$ — a set of variables $\mathbf{X} = \{X_1, \ldots, X_T\}$, each with a domain $\mathbf{D} = \{D_1, \ldots, D_T\}$ of possible discrete values, and a set of constraints that specify the relations between the variables $\mathbf{C} = \{C_1, \ldots, C_K\}$ (Rossi et al., 2006). The goal considered here is to find all assignments of values to the variables that satisfies every constraint. Constrained optimization problems can be also treated as CSPs by converting an additional objective function to maximize or minimize to a soft constraint. Among many learning based approaches, a CSP can be casted as a RL task. In a Markov decision process, the action $\boldsymbol{a}$ corresponds to assigning a set of values to the variables $(x_1, \ldots, x_T)$ where $x_k \in D_k$, and the state $\boldsymbol{s}$ to the current assignment[1], and the reward $R$ to the avoidance of penalties for violating constraints. The policy $\pi$ or the value function $V$ is then optimized to maximize the expected reward by interacting with the environment. The reward function is a crucial component of RL that directly influences the agent's behavior and learning, and for instance, it can be defined as a weighted sum of the degree of satisfying each constraint. Various RL methods have been successfully applied to CSPs with improvement of the search performance (Nareyek, 2004; Xu et al., 2009; Bello et al., 2016), but most of them are limited to well defined static problems and are not directly applicable to address dynamics in the problem.

### 2.2 Unconditional policy generator

An interesting RL algorithm applied to analog circuit design was introduced by Lee & Oliehoek (2020). As an instance of a CSP, its goal is to search for optimal circuit parameters that meet all design constraints imposed on performance metrics. To tackle the problem, the stochastic policy generator is combined with the reward model to take advantage of sample-efficient Dyna-style RL. However, for this type of CSPs where the input variables are statistically independent, each variable is represented by its own distribution that is the output of an independent softmax classifier found in Figure 1, and together these form a policy that is a product distribution. This leads to a simple architecture based on the feedforward network without involving any recurrent units often used for the problems with sequential variables (Zoph & Le, 2016). Furthermore, in order to allow flexible distributions for each variable, the policy network learns a probabilistic distribution of all feasible solutions by mapping a known noise distribution to it, which resembles a generator in unconditional GANs.

### 2.3 Related works

GANs, composed of a generator and a discriminator, employ an adversarial process where these two compete in a minimax game (Goodfellow et al., 2014). The generator aims to produce realistic data samples that can deceive the discriminator, which simultaneously attempts to accurately distinguish between real and generated data. Though developed as distinct branches, potential connection and synergy between RL and GANs have been increasingly recognized and explored by both communities. Pfau & Vinyals (2016) drew the formal connection between GANs and actor-critic methods in RL with insights into a unified theoretical perspective. Ho & Ermon (2016) proposed a GAN-like algorithm for imitation learning to recover a policy that matches the expert demonstrations, and Finn et al. (2016) also showed a mathematical equivalence between maximum entropy inverse RL and GANs. Conversely, RL techniques were exploited to overcome

---

[1]In a strict sense, this RL formulation is stateless because there is no time component in it.

training challenges inherent to GANs. For example, SeqGAN (Yu et al., 2017) used the policy gradient method to train the generator for non-differentiable sequence generation.

## 3 Conditional policy generator

A direct parallel between the policy in RL and the generator in GANs allows us to incorporate recent advances in GANs into the architecture design of the policy network. One straightforward application is to draw inspiration from conditional GANs. Conditional GANs are an extension of vanilla GANs that controls data generation conditioned on additional information (Mirza & Osindero, 2014). Conditions, used as input with random noise, can take various forms including class labels which are also considered in this work, attribute vectors, text descriptions, or other modality data (Mirza & Osindero, 2014; Perarnau et al., 2016; Reed et al., 2016; Isola et al., 2017). Since they serve as certain additional criteria imposed on the GAN generator output which corresponds to input variables of CSPs in case of the policy generator, conditions can be viewed in the context of CSPs as dynamic constraints that are subject to change during the solving process (Heyrani Nobari et al., 2021). Together with standard static constraints that are used in the reward formulation, they form a dynamic CSP that extends the traditional static CSP framework to accommodate dynamics in the problem structure. There are a variety of subsets in the realm of dynamic CSPs, and particularly conditional CSPs (Sabin & Freuder, 1999) can be well suited to our approach. Conditional CSPs, which dynamically activate or deactivate constraints and variables depending on predefined conditions, similarly incorporate conditional logic in their framework.

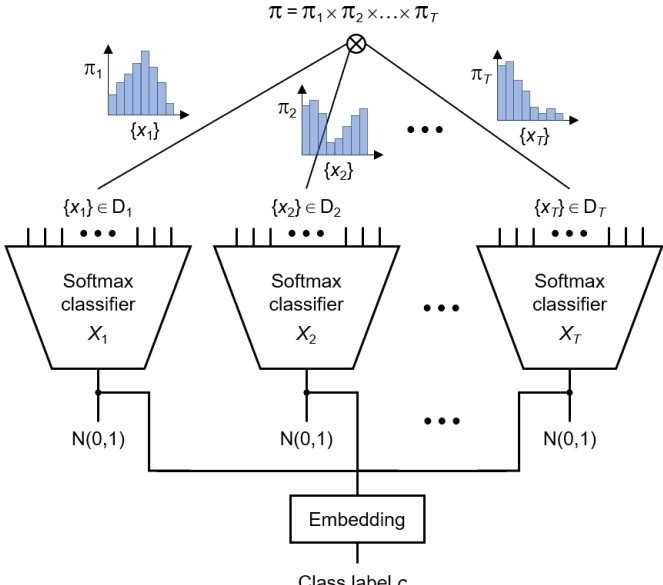

Figure 1: One simple implementation of a conditional policy generator.

Figure 1 shows the architecture of a simple class conditional policy generator to solve a dynamic CSP $\langle \mathbf{X}, \mathbf{D}, \mathbf{C_s}, \mathbf{C_d} \rangle$. It is basically identical to the unconditional policy generator except that the input layer is augmented with additional class labels related to dynamic constraints. The generator consists of a set of independent feedforward networks with the softmax output layers, each of which is responsible for generating a probability distribution of each variable $\mathbf{X}$ over the discrete domain $\mathbf{D}$. The embedding vector of a class label $c$ is concatenated with the random noise vector $\boldsymbol{z}$ sampled from, e.g., the normal distribution $\mathcal{N}(\mathbf{0}, \mathbf{1})$ and fed into each network to control the generator output. The conditional policy generator is trained using a combination of RL and supervised learning. In RL, the standard static constraints $\mathbf{C_s}$ are used to define the reward function as before, which guides the generator to assign high probabilities to solutions satisfying them based on the policy gradient method. In supervised learning, the dynamic constraints $\mathbf{C_d} = \{C_1, \ldots, C_L\}$

are mapped to different classes that can be simply labeled as $c_i = i$ where $i \in \{1, \ldots, L\}$ before applying the embedding, and the generator is trained to maximize the likelihood of classifying the generator output to satisfy a given condition.[2] However, the proposed network design does not require auxiliary discriminators to this end, which makes it distinct from the conditional GAN architecture. Because the generator produces a categorical distribution for each variable that is considered as independent, we can obtain a probability map of the entire input space (i.e., the action space $\mathcal{A}$) for every possible value assignment. This allows us to compute the likelihood of correctly classifying the generated action $\boldsymbol{a}$ given a class by simply summing over the probabilities of all the actions inside the solution subregion $\Omega$ satisfying the given activated condition. The overall training loss can be expressed as

$$
\begin{aligned}
\mathcal{L}(\theta) &= \mathcal{L}_{\mathrm{PG}}(\theta) + \mathcal{L}_{\mathrm{ENT}}(\theta) + \mathcal{L}_{\mathrm{NLL}}(\theta) \\
&= -\frac{1}{N} \sum_{i=1}^{N} \left[ (R_i - b) \log \pi_\theta(\boldsymbol{a}_i | \boldsymbol{z}_i, c_i) + \alpha \mathcal{H}(\pi_\theta(\cdot | \boldsymbol{z}_i, c_i)) + \beta \sum_{\tilde{\boldsymbol{a}} \in \Omega_i} \log \pi_\theta(\tilde{\boldsymbol{a}} | \boldsymbol{z}_i, c_i) \right]
\end{aligned}
\tag{1}
$$

where the entropy $\mathcal{H}(\pi_\theta(\cdot | \boldsymbol{z}_i, c_i))$ and the policy $\pi_\theta(\boldsymbol{a}_i | \boldsymbol{z}_i, c_i)$ are given by

$$
\mathcal{H}(\pi_\theta(\cdot | \boldsymbol{z}_i, c_i)) = -\sum_{\tilde{\boldsymbol{a}} \in \mathcal{A}} \pi_\theta(\tilde{\boldsymbol{a}} | \boldsymbol{z}_i, c_i) \log \pi_\theta(\tilde{\boldsymbol{a}} | \boldsymbol{z}_i, c_i),
\tag{2}
$$

$$
\pi_\theta(\boldsymbol{a}_i | \boldsymbol{z}_i, c_i) = \prod_{t=1}^{T} \pi_{t,\theta}(a_{t,i} | z_{t,i}, c_i).
\tag{3}
$$

Here $\theta$ is the parameters of the generator to be optimized, $N$ the minibatch size, and $T$ the number of input variables. In Equation 1, $\mathcal{L}_{\mathrm{PG}}(\theta)$ represents the REINFORCE algorithm (Williams, 1992) with the baseline $b$ to reduce the variance of the estimate, $\mathcal{L}_{\mathrm{ENT}}(\theta)$ the entropy regularization added to encourage exploration (Mnih et al., 2016), and $\mathcal{L}_{\mathrm{NLL}}(\theta)$ the negative log-likelihood objective for the classification task. The hyperparameters $\alpha$ and $\beta$ are used to balance the contributions of the three loss components, and especially $\beta$ is set to gradually increase from zero during the training because the RL loss is very noisy in the beginning. The detailed training procedure is described in Algorithm 1.

---

**Algorithm 1** Training of proposed conditional policy generator for $\langle \mathbf{X}, \mathbf{D}, \mathbf{C_s}, \mathbf{C_d} \rangle$.

**Require:**
    Define reward function $R$ using static constraints $\mathbf{C_s}$
    Define class labels and solution subregions $\{c_i, \Omega_i\}_{i=1}^{L}$ using dynamic constraints $\mathbf{C_d}$

  1: Initialize generator $\pi_\theta$ with random weights $\theta$
  2: **for** number of training iterations **do**
  3:     **for** $i = 1$ to $N$ **do**
  4:         Sample $\boldsymbol{z}_i \sim \mathcal{N}(\mathbf{0}, \mathbf{1})$
  5:         Sample $c_i$ randomly from $\{c_1, \ldots, c_L\}$
  6:         Sample a vector of $T$ actions $\boldsymbol{a}_i \sim \pi_\theta(\cdot | \boldsymbol{z}_i, c_i)$
  7:         Evaluate reward $R_i = R(\mathbf{X} = \boldsymbol{a}_i)$ where variables $\mathbf{X}$ are bounded by domains $\mathbf{D}$
  8:         Use entropy regularized policy gradient method to compute $\mathcal{L}_{\mathrm{PG}}(\theta)$, $\mathcal{L}_{\mathrm{ENT}}(\theta)$
  9:         Use negative log-likelihood to compute $\mathcal{L}_{\mathrm{NLL}}(\theta)$ based on $\Omega_i$
10:     **end for**
11:     Compute total loss $\mathcal{L}(\theta)$ in Equation 1
12:     Update $\theta$ using gradient descent $\theta \leftarrow \theta - \eta \nabla_\theta \mathcal{L}(\theta)$           $\triangleright$ $\eta$: learning rate
13: **end for**

---

[2]These conditions are supposed to have disjoint solution regions in the input space.

## 4    Experiments

In this section, we present a proof-of-principle experiment to empirically evaluate the performance of the proposed algorithm on a simple multi-modal CSP benchmark. We start with the unconditional policy generator to address the static CSP, and then we extend it to the conditional policy generator by modifying the problem to include dynamic constraints for comparison.

### 4.1    Unconditional case

The test problem, which is called a Synt-3D function as found in Hakhamaneshi et al. (2020), consists of three variables $\mathbf{x} = \{x_1, x_2, x_3\}$, each with a domain of $[-5, 5]$ discretized with 100 possible values. The associated evaluation function is given by $f_{\text{test}}(\mathbf{x}) = \frac{1}{3} \sum_{i=1}^{3} (\frac{1}{4} x_i^4 - 2x_i^2 + 5)$. For simplicity, the goal is to find all value assignments that satisfy a single static constraint of $f_{\text{test}}(\mathbf{x}) < 1.2$. The reward function is expressed as $R = \min\left(\frac{1.2 - f_{\text{test}}(\mathbf{x})}{1.2 + f_{\text{test}}(\mathbf{x})}, 0\right)$. As depicted in Figure 3 (a), this problem has a multi-modal solution space in which each solution mode is formed as a cloud of 248 solution points centered at minima of $(\pm 2, \pm 2, \pm 2)$ within eight different octants in the 3D Cartesian coordinate system. The unconditional policy network is implemented using 3 independent cells of the softmax classifier with 100 output units. The input noise vector per cell is sampled from the Gaussian noise of 64 dimensions, and the generator is trained based on the entropy regularized REINFORCE update rule (i.e., the first two terms in Equation 1) to learn about the action space from the noise prior for generalization.

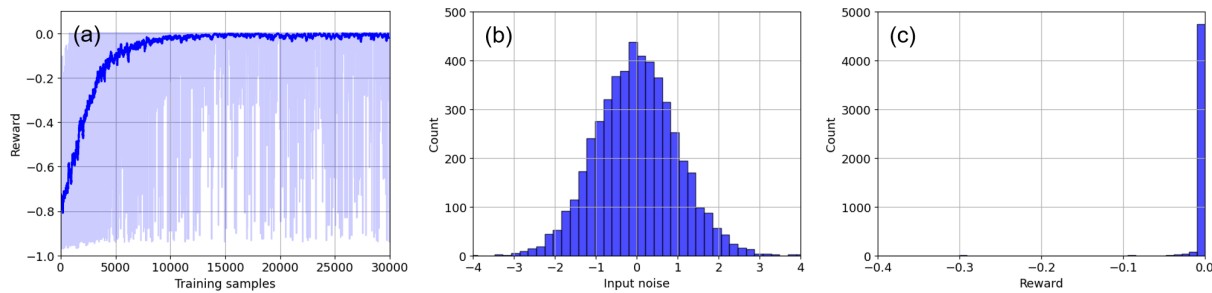

Figure 2: (a) Reward trajectory in policy training, (b) Distribution of Gaussian noise input to trained unconditional policy generator, (c) Reward distribution of 5,000 generated actions from input noise.

Figure 2 (a) shows a noisy reward trajectory while training is carried out over 30,000 iterations. Starting from a negative penalty score, the mean reward averaged over 100 neighbors converges to zero, which means satisfying the constraint, around 10,000 samples as the policy improves. For verification, this trained policy is evaluated by sampling 5,000 actions using the Gaussian noise source as shown in Figure 2 (b), and the reward distribution of the generated actions are presented in Figure 2 (c) which are concentrated mostly near a reward of zero. This represents that the policy generator learns to map the input noise to a distribution of optimal actions solving the problem, similar to the GAN generator.In Figure 3 (b), these output actions are also plotted in the 3D Cartesian coordinates to compare with Figure 3 (a), and it is evident that all the solution modes in the different octants are evenly discovered well by the policy generator without the mode collapse which is often problematic for the GAN generator. If necessary, the sampling efficiency can be significantly improved by incorporating the learnable reward model and applying the Dyna-style learning framework (Lee & Oliehoek, 2020). For the baseline comparison, we also evaluate the generalized autoregressive cross-entropy method (GACEM) algorithm (Hakhamaneshi et al., 2020) to solve the same problem. It uses a masked auto-regressive neural network and RL training to learn diverse solution distributions more effectively than traditional cross-entropy methods based on simple Gaussian models. It can be trained either on-policy, updating with samples from the current distribution for better scalability, or off-policy, using a replay buffer to maintain solution diversity in lower dimensions. We train both versions of GACEM with 30,000 samples same as before, and the results of 5,000 evaluation samples from the trained policies are shown in Figures 3 (c) and (d). It can be seen that the on-policy GACEM fails to discover

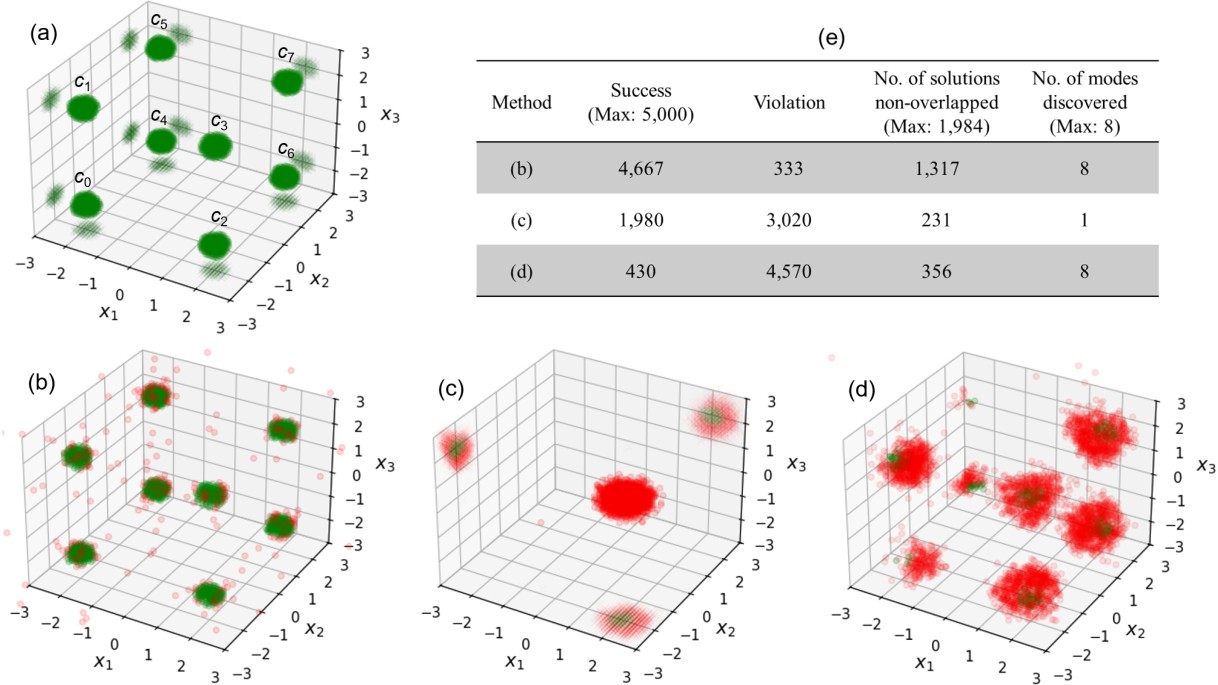

Figure 3: Distributions of 5,000 evaluation samples (marked with 'o') in action space including its projection (marked with 'x', if any) onto coordinate planes for baseline comparison: (a) Exact multi-modal solution of Synt-3D problem, (b) Our approach, (c) On-policy GACEM, (d) Off-policy GACEM, (e) Performance metrics comparison. Green and red symbols indicate ones that succeed in and violate the recovery of solution, respectively.

diverse solution modes, while the off-policy GACEM finds them but with much less uniformity compared to our approach. Figure 3 (e) summarizes the performance metrics of three methods, which shows that our unconditional policy generator outperforms both GACEM methods in terms of solution diversity, prediction accuracy, and mode coverage.

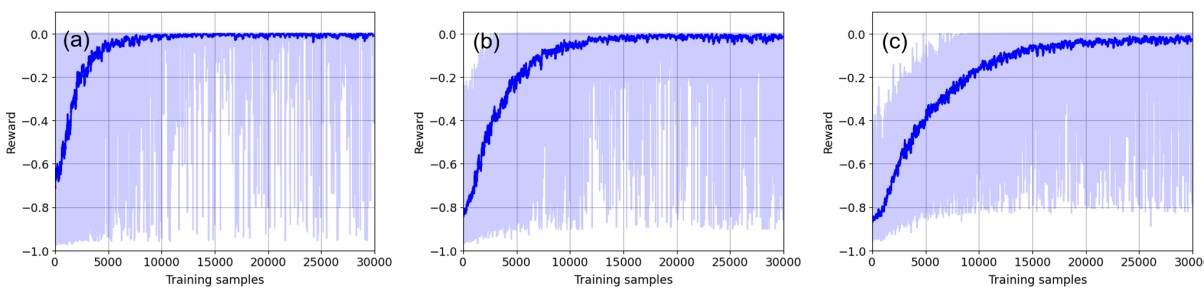

Figure 4: Comparison of reward trajectories of unconditional policy training in (a) Synt-2D, (b) Synt-5D, and (c) Synt-10D problems.

Another advantage of this approach is expected to shine when dealing with high dimensional problems. The simple neural architecture based on feedforward network not only allows parallel computation, but also each cell in the generator enables us to *divide and conquer* independent variables one by one. Figure 4 shows the reward trajectories of Synt-2D, Synt-5D, and Synt-10D for comparison. Regardless of the problem dimensionality, it is interesting to note that they all exhibit similar learning convergence by marginalizing and optimizing each variable independently. Although the number of training samples required for the Synt-2D

problem is even comparable to the full grid search case, it does not exponentially increase with the problem dimensionality, hence effectively mitigating the curse of dimensionality.

## 4.2 Conditional case

In order to demonstrate the feasibility of the proposed approach to solving dynamic CSPs, we modify the Synt-3D problem used in the previous section to include eight additional dynamic constraints dictating whether the assignment of values belongs to each octant space defined by the signs of three coordinate variables. The goal in this section is to find one of eight solution modes depending on the conditional input selected from $\{c_0, \ldots, c_7\}$ classes where $c_i$ corresponds to occupancy in the $i$-th octant space as shown in Figure 3 (a). The class label $c_i = i$ is embedded and concatenated with the noise vector as input to the policy generator, and the training is performed based on Algorithm 1.

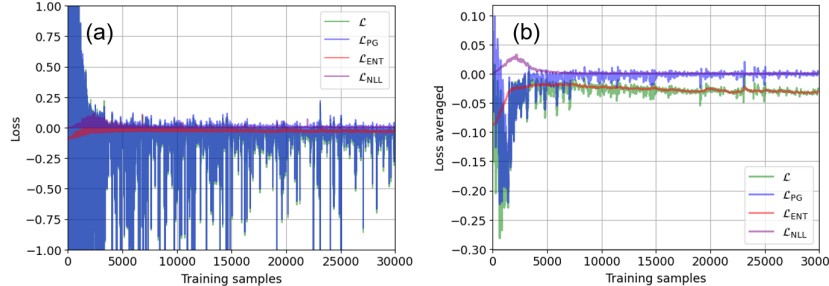

Figure 5: Trajectories of (a) loss and (b) its moving average over 100 neighboring values during the training of conditional policy generator.

Note that the reward only reflects the static constraint as before, and so the learning progress of the policy satisfying both the static and dynamic constraints can be traced better via the loss trajectories as shown in Figure 5. In the beginning, the policy update uses the noisy policy gradient loss with the relatively high entropy bonus to better explore the action space, and then the negative log-likelihood loss for classification gradually comes into play together as $\beta$ in Equation 1 increases from zero. Both loss components approach zero similarly in less than 10,000 samples, and the entropy loss is saturated to a lower level as the policy improves and becomes more certain about solution to the problem. The trained policy is evaluated by generating 1,000 action samples from the noise prior conditioned on different class input, and the class-conditioned sample plots are presented in Figure 6. The boundary-confusion analysis including the multi-class confusion matrix is also performed, and it can be seen that the generated actions per class are well aligned with the corresponding octant space and correctly reproduce the solution mode within it, which indicates that the conditional policy generator is able to learn to solve the dynamic CSP by effectively searching the entire solution space satisfying both static and dynamic constraints.

## 5 Limitations and future work

The proposed method has several limitations that need to be addressed in future work. First, the current implementation assumes that the input variables are statistically independent, which may not hold in many real-world problems with correlated variables. Extending our approach to handle such dependencies would require more sophisticated network architectures, such as incorporating attention mechanisms (Vaswani et al., 2017), to enhance the expressiveness of the policy generator. Second, while the feasibility of our approach has been demonstrated using a simple benchmark problem with low dimensionality and discrete domains, its performance on more complex and high-dimensional problems remains to be evaluated. This includes conducting extensive experiments on various benchmark datasets and comparing our method with existing state-of-the-art methods in terms of solution quality, computational efficiency, and scalability. Although one simple implementation of the proposed architecture that is effective for a low dimensional problem is presented in this work, the tractability of summing class-likelihood over feasible solutions given in Equation 1 could be an issue in higher dimensions. One alternative to avoid this challenge is to add an auxiliary

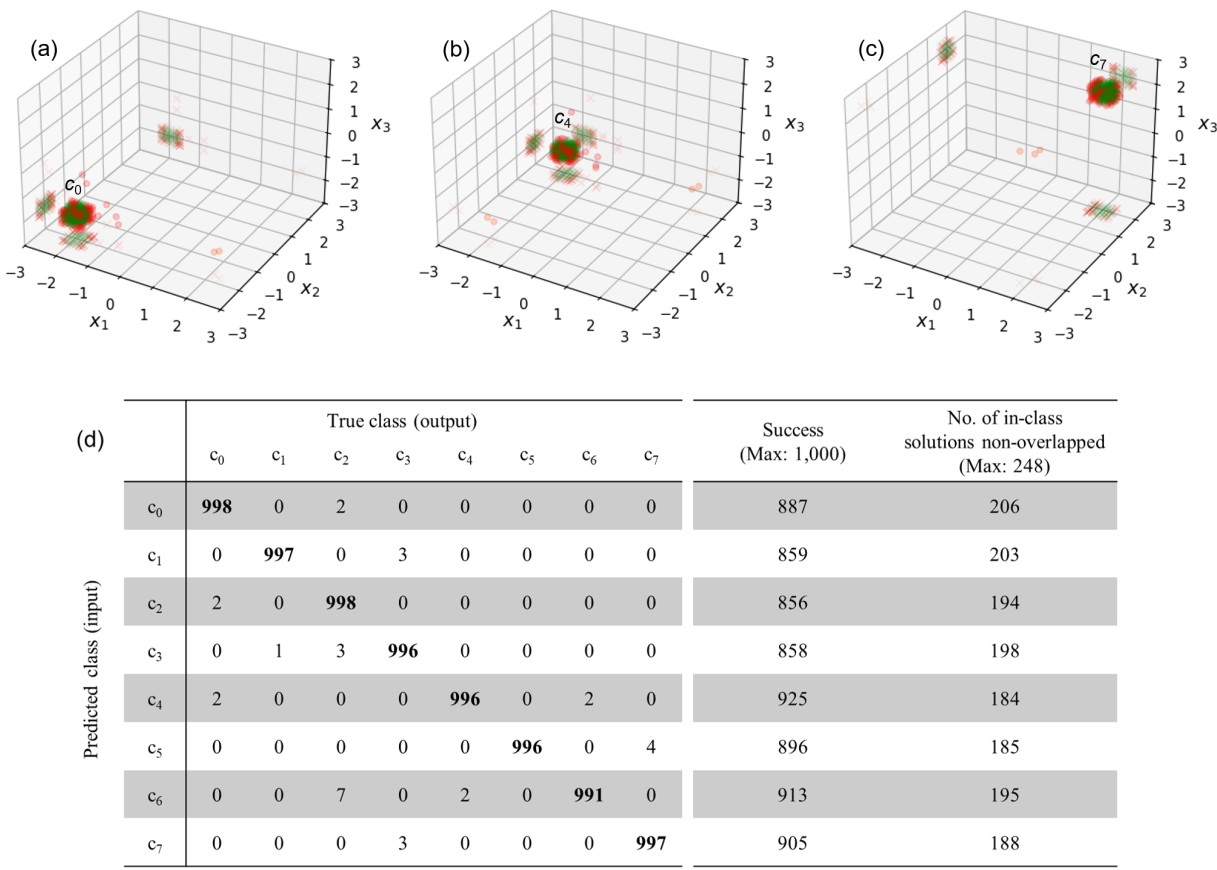

| | | | | | True class (output) | | | | | | Success (Max: 1,000) | No. of in-class solutions non-overlapped (Max: 248) |
|---|---|---|---|---|---|---|---|---|---|---|---|---|
| | | $c_0$ | $c_1$ | $c_2$ | $c_3$ | $c_4$ | $c_5$ | $c_6$ | $c_7$ | | | |
| | $c_0$ | **998** | 0 | 2 | 0 | 0 | 0 | 0 | 0 | | 887 | 206 |
| | $c_1$ | 0 | **997** | 0 | 3 | 0 | 0 | 0 | 0 | | 859 | 203 |
| | $c_2$ | 2 | 0 | **998** | 0 | 0 | 0 | 0 | 0 | | 856 | 194 |
| Predicted class (input) | $c_3$ | 0 | 1 | 3 | **996** | 0 | 0 | 0 | 0 | | 858 | 198 |
| | $c_4$ | 2 | 0 | 0 | 0 | **996** | 0 | 2 | 0 | | 925 | 184 |
| | $c_5$ | 0 | 0 | 0 | 0 | 0 | **996** | 0 | 4 | | 896 | 185 |
| | $c_6$ | 0 | 0 | 7 | 0 | 2 | 0 | **991** | 0 | | 913 | 195 |
| | $c_7$ | 0 | 0 | 0 | 3 | 0 | 0 | 0 | **997** | | 905 | 188 |

Figure 6: Distributions of 1,000 generated samples (marked with 'o') in action space including its projection (marked with 'x') onto coordinate planes for evaluation of trained conditional policy generator with conditional input of (a) class $c_0$, (b) class $c_4$, or (c) class $c_7$. Green and red symbols indicate ones that succeed in and violate the recovery of solution, respectively.(d) Performance metrics including multi-class confusion matrix.

classifier in the architecture. It is then trained together with the policy generator in the same way so that the classifier discriminates the generated action to its corresponding class depending on a given conditioning label input with noise. Finally, the similarity between the policy generator and the GAN generator opens up an interesting avenue for future research, where we may leverage other advanced GAN techniques to enhance the performance of CSP solvers and develop more efficient, robust, and scalable algorithms.

## 6 Conclusions

We propose a new approach to addressing dynamic constraint satisfaction and optimization problems by introducing a class conditional policy generator based on the RL framework by taking inspiration from the idea of conditional GANs. By assuming that the dynamic CSP consists of both static and dynamic constraints for generalization as well as independent variables, the proposed algorithm is designed to learn a probabilistic distribution of solutions in the action space from a noise prior using the entropy regularized policy gradient method to satisfy static constraints, while simultaneously adapting to dynamic constraints through the conditioning class labels based on the maximum likelihood estimation. We demonstrate the feasibility of our approach with a simple multi-modal CSP benchmark problem, and show that the conditional policy generator can effectively learn to solve the problem by exploring the solution space.

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
