# OpenReview forum: "Conditional Policy Generator for Dynamic Constraint Satisfaction and Optimization"
_TMLR — Rejected by TMLR_

### Review · Reviewer_K93q · 2025-08-09

**Summary Of Contributions:**

The paper considers the dynamic constraint satisfaction problem where some constraints are static (used only in a reward) and others are “dynamic” (encoded as class labels). The paper assumes variables are individually independent and models the policy as a product of per-variable softmax outputs with gaussian noise inputs. The training procedure consists of minimizing a combination of loss: (i) vanilla policy gradient loss on the reward from static constraints, (ii) an entropy bonus, and (iii) a classification term derived by summing log probabilities of all actions in the class-specific solution subregion. Experiments are performed on a toy Synt-3D function domain discretized to 100 values per variable, with unconditional (static) and conditional (octant-label) variants.

**Audience:**

No

**Audience Explanation:**

- The paper's introduction mentions an important problem but the rest of the paper doesn't match the description.

**Claims And Evidence:**

No

**Claims Explanation:**

- Based on the introduction, it seemed to me that the dynamic aspect would be temporal but it doesn't seem to be the case. The RL formulation has no time component. The dynamic part mostly corresponds to conditioning on labels.

- The method assumes variables are independent and the policy factorizes across variables. This significantly simplifies the problem setup.

- The experimental evaluation is quite narrow and limited to a toy task of discretized 3 variable function. There are no baselines to compare with.

- The "supposed" connection to conditional GAN is mostly superficial. There is no discriminator used, so beyond concatenating a label to the input, it’s unclear what is imported from cGANs.

- Key hyperparameter details are missing. Algorithm 1 is mostly high-level and no description of  values for parameters like alpha or beta is provided.

**Requested Changes:**

- Since the method itself is straightforward, it would have been great to see some application on a real world problem.

---

> ### Author Response · Authors · 2025-09-02
> **Response to Reviewer K93q**
>
> ## Response to Requested Changes
>
> > Since the method itself is straightforward, it would have been great to see some application on a real world problem.
>
> The goal of this paper is to provide proof of concept results for a new RL-based approach to conditional CSPs. We agree that the empirical evaluation is limited, but think they are sufficient to spark interest in this new approach. Unfortunately we have no suitable application results on a real world problem at this point. Instead we have included a new section on limitations and future work to address this point.
>
> &nbsp;
> &nbsp;
>
> ## Self-response to other comments
>
> We thank the reviewer for all the valuable and helpful feedback, but there seems to be a couple of misunderstandings that we would like to clarify in the following.
>
> > Based on the introduction, it seemed to me that the dynamic aspect would be temporal but it doesn't seem to be the case. The RL formulation has no time component. The dynamic part mostly corresponds to conditioning on labels.
>
> The dynamic aspect can be attributed to many things, such as variation over time (i.e, temporal), conditional change, or even uncertainty or incomplete knowledge in the problem components. In this work we focus on tackling the problem which is subject to conditional change.
>
> > The method assumes variables are independent and the policy factorizes across variables. This significantly simplifies the problem setup.
>
> We acknowledge the limitation of our method by assuming independent variables. This problem setup was described in detail in the previous work [1]. Some part of optimization problems in engineering and science have variables that are only loosely interacting and can be regarded as statistically independent, such as the analog circuit sizing problem in [1]. Actually we attempted to solve it with RL, which in turn led to a simple policy architecture derived from the RNN policy controller for neural architecture search [2]. We agree that our approach needs to be expanded for sure, and it has been discussed in a new section.
>
> > The "supposed" connection to conditional GAN is mostly superficial. There is no discriminator used, so beyond concatenating a label to the input, it’s unclear what is imported from cGANs.
>
> We are interested in the similarity between our policy network in RL and the generator in cGANs. For example, the policy also learns mapping from the noise input to the target distribution, and conditioning labels used with the noise control the output. In addition, the class likelihood of satisfying a given condition is calculated in the same way, but there are no discriminative counterparts in our proposed architecture because it can be directly computed using the output of the policy network made of softmax classifiers. We agree that a defining feature of GANs is not present in a strict sense, but retain this terminology since it makes it easier to explain and to credit our inspiration.
>
> &nbsp;
> &nbsp;
> &nbsp;
>
> ******
>
> [1] Wook Lee and Frans A Oliehoek. Analog circuit design with dyna-style reinforcement learning. In NeurIPS 2020 Workshop: Machine Learning for Engineering Modeling, Simulation, and Design, 2020.
>
> [2] Barret Zoph and Quoc V Le. Neural architecture search with reinforcement learning. arXiv preprint arXiv:1611.01578, 2016

---

### Review · Reviewer_ajbn · 2025-08-16

**Summary Of Contributions:**

The authors of this paper tackle Constraint Satisfaction Problems (CSPs) with evolving constraints, using a combination of Reinforcement Learning (RL) and generative modeling. They consider the framework of a CSP with both static and dynamic constraints, the latter being dependent on a discrete class variable. To solve it, they rely on a source of random noise that passes through an encoder, similar to Generative Adversarial Networks (GANs). Based on the resulting encoding, each decision variable is independently decoded with a softmax activation over its domain. The loss minimized during training combines a policy gradient term for the static constraints, a likelihood term for the dynamic constraints, and an entropic regularization. A proof-of-principle experiment in low dimension illustrates the method.

**Additional Comments:**

Minor remarks:

- The assumption of variables being "independent of each other" is never clearly defined or justified. If the variables were truly independent, the CSP would be trivial.
- When defining the reward, how is "the degree of satisfying each constraint measured"?
- The class labels, solution subregions and constraints are all named C, which can cause confusion for the readers.
- On figure 2, it is not necessary to plot the histogram of a standard Gaussian.
- What does it mean to "average over 100 neighbors" when evaluating the loss? How are the neighbors picked?

**Audience:**

No

**Audience Explanation:**

- The proposed method is not adequately compared to the state of the art on dynamic combinatorial problems with contextual information [1]. In fact, the related works section barely mentions the rich literature on the interplay between constraint satisfaction and machine learning, since it is solely focused on the interplay between GANs and RL.
- The CSP is modeled as a Markov Decision Process but each action sets the entirety of the decision variables, completely obviating the sequential aspect. In many real-world problems like scheduling (which are mentioned in the introduction), some decisions are made only once and cannot be revisited later in time. Additionally, this makes the Markovian model less relevant.
- No practical examples are provided of CSPs where the dynamic aspects are entirely captured by one categorical variable.

---

[1] U. Sadana, A. Chenreddy, E. Delage, A. Forel, E. Frejinger, and T. Vidal, “A survey of contextual optimization methods for decision-making under uncertainty,” European Journal of Operational Research, Mar. 2024, doi: 10.1016/j.ejor.2024.03.020.

**Broader Impact Concerns:**

No broader impact concerns.

**Claims And Evidence:**

No

**Claims Explanation:**

- The learning algorithm introduced here seems limited to low-dimensional problems. Indeed, the third term of the loss in Equation (1) sums over all the actions (= complete solutions) in the subregion $\mathcal{C}_i$ satisfying the current constraints, a set which can be exponentially large as a function of the dimension. In other words, the "probability map of the entire input space (i.e. the action space $\mathcal{A}$)" promised by the authors becomes computationally intractable as soon as the ambient dimension grows.
- The experiments are very narrow in scope, only considering one type of objective function (which is continuous at heart) and remaning in low dimension (from a minimum of 2 variables to a maximum of 10 variables). As the authors notice, the instance is so small that "the number of training samples required for the Synt-2D problem is even comparable to the full grid search case". However, their claim that "it does not exponentially increase with the problem dimensionality, hence effectively mitigating the curse of dimensionality" is not founded due to the lack of larger experiments.

**Requested Changes:**

Unfortunately, I do not think this paper can meet the acceptance threshold for TMLR, even after significant revision. Still, here are my main recommendations for improvement:

- Explain how this method would scale to combinatorial problems with thousands of variables.
- Provide more varied and large-scale experiments.
- Insert the suggested approach inside the wider literature on contextual and multi-stage optimization.
- Discuss how to make it truly sequential, instead of redefining the whole decision at every time step.

---

> ### Author Response · Authors · 2025-09-02
> **Response to Reviewer ajbn**
>
> ## Response to Requested Changes
>
> > Explain how this method would scale to combinatorial problems with thousands of variables.
>
> Since the policy network is quite as simple as a combination of each softmax classifier per variable, it could be straightforward to expand it with thousands of variables and train the policy based on this feedforward network. We think this point is already mentioned in Section 4.1, “The simple neural architecture based on feedforward network not only allows parallel computation …”
>
> > Provide more varied and large-scale experiments.
>
> Unfortunately we have no suitable results on more varied and large-scale experiments at this point even though we think its application can be straightforward. The goal of this paper is to provide proof of concept results for a new RL-based approach to conditional CSPs. We agree that the empirical evaluation is limited, but think they are sufficient to spark interest in this new approach. Alternatively we have included a new section on limitations and future work to address this point.
>
> > Insert the suggested approach inside the wider literature on contextual and multi-stage optimization.
>
> We have included the survey literature [1] in the reference. However note that the problem of our interest involves dynamics of conditional change in the problem setting, but does not consider the uncertainty or incomplete knowledge about it at all.
>
> > Discuss how to make it truly sequential, instead of redefining the whole decision at every time step.
>
> We are not completely certain what the reviewer means by this. We proposed an RL based solution for a class of optimization problems. So the solution approach is sequential, while the problem is not necessarily sequential (even though we allow for dynamic constraints that could vary on an ad-hoc basis).
>
> &nbsp;
> &nbsp;
>
> ## Response to Additional Comments
>
> > The assumption of variables being "independent of each other" is never clearly defined or justified. If the variables were truly independent, the CSP would be trivial.
>
> We think analog circuit sizing problem in [2] is a good example where the variables can be viewed as statistically independent. In case of stronger dependencies, such an RNN (or transformer) could be incorporated in our approach too.
>
> > When defining the reward, how is "the degree of satisfying each constraint measured"?
>
> The degree of constraint satisfaction is obtained from the evaluation function f_test(x), and the reward is defined in Section 4.1. You can also refer to Equation 1 in [2].
>
> > The class labels, solution subregions and constraints are all named C, which can cause confusion for the readers.
>
> We have changed the notation of solution subregions.
>
> > On figure 2, it is not necessary to plot the histogram of a standard Gaussian.
>
> We agree that it does not mean much by itself. However we would like to keep it as a pair with (c) to illustrate the policy mapping from the noise to target distributions.
>
> > What does it mean to "average over 100 neighbors" when evaluating the loss? How are the neighbors picked?
>
> Since the loss is quite noisy, we simply applies a moving average with 100 consecutive neighboring values for the display purpose. We have changed the figure caption accordingly.
>
> &nbsp;
> &nbsp;
> &nbsp;
>
> ******
>
> [1] U. Sadana, A. Chenreddy, E. Delage, A. Forel, E. Frejinger, and T. Vidal, A survey of contextual optimization methods for decision-making under uncertainty. European Journal of Operational Research, 2024.
>
> [2] Wook Lee and Frans A Oliehoek. Analog circuit design with dyna-style reinforcement learning. In NeurIPS 2020 Workshop: Machine Learning for Engineering Modeling, Simulation, and Design, 2020.

---

> > ### Comment · Reviewer_ajbn · 2025-09-08
> >
> > > Since the policy network is quite as simple as a combination of each softmax classifier per variable, it could be straightforward to expand it with thousands of variables and train the policy based on this feedforward network. We think this point is already mentioned in Section 4.1, “The simple neural architecture based on feedforward network not only allows parallel computation …”
> >
> > It does not seem so straightforward to me. The key limitation is not the architecture itself but the loss, more specifically summing over the set $\Omega_i$ in equation (1), which will contain exponentially many solutions. Unless I am mistaken, your work does not address this crucial barrier to scaling.
> >
> > > We are not completely certain what the reviewer means by this. We proposed an RL based solution for a class of optimization problems. So the solution approach is sequential, while the problem is not necessarily sequential (even though we allow for dynamic constraints that could vary on an ad-hoc basis).
> >
> > The meaning of my remark is that you formulate the learning problem as an MDP, but there are no Markovian dependencies since the action $a$ contains every single decision variable. In typical RL approaches to combinatorial problems, the solution is constructed little by little (e.g. customer by customer or route by route in a VRP), which means that the notion of state makes sense. Here, your method does not leverage state (each sample is independent from the last), which makes me wonder why RL is relevant at all. This looks more like a bandit-related setting.
> >
> > > We think analog circuit sizing problem in our previous work [2] is a good example where the variables can be viewed as statistically independent.
> >
> > Please do not unblind the peer review by referring to your previous work explicitly.

---

> > > ### Author Response · Authors · 2025-09-09
> > > **Response to Reviewer ajbn**
> > >
> > > > It does not seem so straightforward to me. The key limitation is not the architecture itself but the loss, more specifically summing over the set in equation (1), which will contain exponentially many solutions. Unless I am mistaken, your work does not address this crucial barrier to scaling.
> > >
> > > Although we demonstrated one simple implementation of the proposed architecture that is effective in a low dimensional problem, we agree that tractability of directly summing class-likelihood over feasible regions in equation (1) could be an issue in higher dimensions. To avoid this challenge, we think one alternative, which could be more standard, is to add an auxiliary classifier in the architecture. We then train it with the policy generator together in the same way so that the classifier discriminates the generated sample to its corresponding class depending on a given conditioning label input with noise. We have rewritten a section on limitations and future work to include ths point.
> > >
> > > > Please do not unblind the peer review by referring to your previous work explicitly.
> > >
> > > Thanks for letting us know. It has been corrected in our original comment.

---

### Review · Reviewer_WNMe · 2025-08-19

**Summary Of Contributions:**

A conditional policy generator is proposed for dynamic constraint satisfaction and optimization. Static constraints are embedded in a reward used for entropy-regularized policy-gradient training, while dynamic constraints are mapped to class labels and handled with a negative log-likelihood objective; the overall loss adds REINFORCE, entropy, and class-likelihood terms. A proof-of-principle experiment on a synthetic multi-modal CSP (Synt-3D) shows that the factorized policy (independent softmax per variable, noise-conditioned) recovers all modes and, in the conditional case, targets octant-specific solution regions. The contribution is positioned as GAN-inspired (conditional input, generator-like mapping from noise) but remains purely RL + supervised without a discriminator.

**Audience:**

Yes

**Audience Explanation:**

Yes, niche interest. The findings would interest readers in RL-for-CSP, conditional generative modeling, and discrete optimization, but broader TMLR appeal remains limited without stronger benchmarks and baselines.

**Claims And Evidence:**

Yes

**Claims Explanation:**

Evidence is clear and accurate for a toy CSP (feasibility shown), but not yet convincing for broader dynamic CSPs or scalability:
- No comparisons to strong baselines or sample-efficiency metrics.
- Tractability of summing class-likelihood over feasible regions in higher dimensions is not analyzed.

**Requested Changes:**

Add rigorous empirics: strong baselines (e.g., CEM/heuristics/RL), quantitative metrics (success/violations, sample efficiency, wall-clock), and ablations (α/β schedules, entropy, noise dim, discretization).

Strengthen conditional results: per-class precision/recall of feasible mass, boundary-confusion analysis, and class-conditioned sample plots.

Address scalability: analyze/approximate the class-likelihood sum over feasible sets (e.g., importance sampling, amortized estimators); report compute/time scaling with dimension.

Notation/ Grammar:

Duplicate word: “the the policy generator” → “the policy generator.”
Construction: “does not only allow parallel computation” → “not only allows parallel computation.”
Gerund: “enables to divide and conquer” → “enables divide-and-conquer” (or “enables us to divide and conquer”).
Idiom: “satisfy to a given condition” → “satisfy a given condition.”

Proofreading is needed.

---

> ### Author Response · Authors · 2025-09-02
> **Response to Reviewer WNMe**
>
> ## Response to Requested Changes
>
> > Add rigorous empirics: strong baselines (e.g., CEM/heuristics/RL), quantitative metrics (success/violations, sample efficiency, wall-clock), and ablations (α/β schedules, entropy, noise dim, discretization).
>
> For the baseline comparison, we have included two evaluation results of modified CEM approach [1] (on-policy & off-policy GACEM in Fig. 3 (c) & (d)) as well as a discussion in Section 4.1. Regarding quantitative metrics, we also have added performance metrics comparison including success/violation counts in Fig. 3 (e).
> In addition, since the minibatch size of one is used in all the experiments, the number of training steps in figures is equal to the number of training samples. So we have changed the x-axis title accordingly in Fig. 2 (a), Fig. 4 (a)-(c), and Fig. 5 (a)-(b). For better sample efficiency, dyna-style RL can be applied [2].
>
> > Strengthen conditional results: per-class precision/recall of feasible mass, boundary-confusion analysis, and class-conditioned sample plots.
>
> We think the class-conditional sample plots are presented in Fig. 6 (a)-(c). We have conducted the boundary-confusion analysis, and added the results including a multi-class confusion matrix in Fig. 6 (d).
>
> > Address scalability: analyze/approximate the class-likelihood sum over feasible sets (e.g., importance sampling, amortized estimators); report compute/time scaling with dimension.
>
> Our experimental evaluation was carried out in a low-dimension problem as a proof of concept. We agree that the class-likelihood sum over feasible solutions could become challenging in higher dimensions and its scalability needs to be examined. We have included a new section on limitations and future work to address this issue.
>
> > Proofreading is needed.
>
> Typos & mistakes have been fixed.
>
> &nbsp;
> &nbsp;
> &nbsp;
>
> ******
>
> [1] Kourosh Hakhamaneshi, Keertana Settaluri, Pieter Abbeel, and Vladimir Stojanovic. GACEM: Generalized autoregressive cross entropy method for multi-modal black box constraint satisfaction. arXiv preprint arXiv:2002.07236, 2020.
>
> [2] Wook Lee and Frans A Oliehoek. Analog circuit design with dyna-style reinforcement learning. In NeurIPS 2020 Workshop: Machine Learning for Engineering Modeling, Simulation, and Design, 2020.

---

### Comment · Reviewer_5ia8 · 2025-08-15
**Request to unassign me from this paper**

Dear Editor,

Earlier another editor (Chicheng Zhang) reached out to me personally to ask me to review a paper in my field (active learning). I agreed but you assigned this paper to me before him. I’m working full time now and have limited time. Therefore, I can’t review multiple papers at this time. Could you please unassign this paper from me so I can be assigned to another paper? Thank you very much!

Best,
Joey

---

### Decision · Action_Editor_Pow3 · 2025-09-18

**Recommendation:** Reject

**Audience:**

No

**Audience Explanation:**

It seems plausible that the method itself is not suited to larger scale CSP problems. In addition, there are already a number of works using RL & RL-adjacent methods to solve CSP or CSP-like problems. This work is neither novel enough nor empirically complex enough to be likely to find a significant audience within TMLR.

**Claims And Evidence:**

No

**Claims Explanation:**

There is a consensus by reviewers that the experiments are too simple to be considered convincing evidence of a method that can tackle general CSP problems.